# Sample Adequacy Control (SAC) Lowers False Negatives and Increases the Quality of Screening: Introduction of “Non-Competitive” SAC for qPCR Assays

**DOI:** 10.3390/diagnostics11071133

**Published:** 2021-06-22

**Authors:** Ivan Brukner, Alex Resendes, Shaun Eintracht, Andreas I. Papadakis, Matthew Oughton

**Affiliations:** 1Lady Davis Institute for Medical Research, Montréal, QC H3T 1E2, Canada; alexresendes27@hotmail.com (A.R.); andreas.papadakis@ladydavis.ca (A.I.P.); 2Faculty of Medicine, McGill University, Montreal, QC H3A 0G4, Canada; shaun.eintracht@mcgill.ca

**Keywords:** sampling, control, assay, qPCR, pandemic, infection control, false negative, clinical, screening, test

## Abstract

Sample Adequacy Control (SAC) has critical analytical, clinical and epidemiological value that increases confidence in a negative test result. The SAC is an integral qPCR assay control, which ensures that all pre-analytical and analytical steps are adequate for accurate testing and reporting. As such, a negative SAC with a negative result on pathogen screen specifies that the result should be reported as inconclusive instead of negative. Despite this, many regulatory approved tests do not incorporate SAC into their assay design. Herein, we emphasize the universal value of SAC and offer for the first time, a simple technical strategy to introduce non-competitive SAC which does not interfere with the limit of detection for the screened pathogen. Integration of SAC can provide key benefits towards identifying, isolating, quarantining and contact tracing infected individuals and in turn can improve worldwide efforts in infection control.

## 1. Introduction

Sample adequacy control (SAC) is a “built-in check” to ensure that the sample is suitable and meets the requirements for testing [1,2,3,4,5,6]. Usually, an endogenous sample-specific (human) gene is co-amplified to assure that adequate sample has been collected. In a clinical setting of medical diagnostics, it is critically important to institute confidence in a negative result, which without SAC cannot be achieved. Reducing false negative (FN) results is of extreme importance for infection control efforts, especially during pandemic times.

Impact of false negatives on infection control: Poor assay sensitivity is directly proportional to the number of false negative results [7]. However, even assays with the most optimal analytical performance can yield FN results: e.g., when sampling is inadequate (below the minimal quantity of biological material collected), or the integrity of the sample is compromised during transportation and/or storage, when the assay is inhibited, or when there is any other pre-analytical manipulation compromising the accuracy of reported results [2,8,9,10,11]. Although these can be largely mitigated by streamlined and effective testing pipelines, the need for sample adequacy controls become critically essential, as testing volumes increase, when sample types become diverse and when self-sampling strategies become incorporated into testing algorithms. For example, during the Covid-19 pandemic, the number of suggested screening sample types has increased. Initially, these were limited to nasopharyngeal swabs, but now have been extended to saliva, mouth gargles, nasal, oral and rectal swabs, as well as self-collected samples [10,12,13,14,15,16,17,18,19,20]. The relative frequency of FN samples is generally high (2–38%) and dependent on multiple variables [7,9,10]. FN screening results for COVID-19 test have detrimental social, economic and health impact, by offsetting the effectiveness of screening efforts and prolonging the pandemic. Removing even a small fraction of FN samples will allow for improvement in infection control measures and a more rapid and safer loosening of restrictions and the recommencement of essential, social and economic activities.

Quality control requirements for screening assays: The MIQE rules are among some of the primary quality requirements to be followed when validating a new screening qPCR assay [21,22,23,24]. These rules also stipulate that quantitative, or semi-quantitative qPCR analysis can be performed by relative quantification with a second biomarker control, equivalent to SAC [22,25].

Pitfalls in qPCR-based diagnosis can reduce the effectiveness of public infection control efforts [16,17,19,26,27,28,29,30]. For example, even though emergency uses of different COVID-19 tests are authorized, recommendations made by CDC (https://www.cdc.gov/coronavirus/2019-ncov/index.html or https://www.cdc.gov/coronavirus/2019-ncov/lab/virus-requests.html, accessed on 6 June 2021) and WHO (https://www.who.int/diagnostics_laboratory/eual/eul_0515_202_00_covid19_coronavirus_real_time_pcr_kit_ifu.pdf, accessed on 6 June 2021) which assure quality of testing must include: (a) control of sampling, (b) control of reaction inhibition, (c) control of the chemical integrity of biomarker (RNA, or DNA, or protein…) and (d) the control of sample processing/extraction. These types of quality controls should not be added externally [31] but should be intrinsically present in the sample [1,3,16,21,22,32]. SAC unifies these quality assurance controls. If pathogen genetic material is not detected and SAC is negative—the result should be declared as invalid, or inconclusive. The absence of SAC usage in screening assays, can be detrimental, as individuals harboring infection and declared “negative” (instead of inconclusive), would not be flagged to be re-tested, would not self-isolate and could potentially spread the infection.

In this work, we emphasize the importance of sample adequacy in medical diagnostics. Sample adequacy is characterized by sufficient quantity and integrity of disease-specific biomarker present in the sample. One approach to establishing a reliable indicator of adequacy is to examine the frequency distribution of the sample-specific biomarker(s) in advance. This statistical approach is arbitrary but demonstrates that not all samples are equal in their biomarker quantity. For example, if a sample-specific biomarker quantity is found to be less than two standard deviations from its mean frequency distribution for a particular sample type, then this sample belongs to a small group of only 2.5% of samples. The rest of the samples (97.5%) contain higher biomarker quantity. The second approach is pathogen/disease specific. In this method, the lowest ratio of disease-specific biomarker vs. sample-specific biomarker must be determined. For example, we never detected nasopharyngeal samples having an Influenza A virus density lower than ~0.0001 per cell (i.e., one genome-equivalent of Influenza A virus in the context of ~ 10,000 human genome equivalents) [1]. To achieve a maximal clinical/epidemiological sensitivity, samples should never have less than ~10,000 human genome equivalents. When the sample contains less than ~10,000 human genome equivalents, the risk of obtaining a false negative result when screening for a pathogen significantly increases. Since the concept of SAC is based on the frequency distributions of biomarkers, choosing a quantitative “threshold” for a sample-specific biomarker might depend on the gravity of the disease (Ebola vs. simple respiratory infection) and the necessity of not missing a case. A major drawback of incorporating SAC in the same reaction compartment is that it decreases disease-specific assay sensitivity by competitive inhibition. Here, we offer, for the first time, a simple technical strategy to introduce non-competitive sample-specific biomarker SAC in the same reaction mix, together with a disease-specific biomarker.

## 2. Methods

Rectal and nasopharyngeal swabs/samples were collected and analyzed in the Microbiology Department at Jewish General Hospital (Montreal, QC, Canada), as previously described [1,3]. Swabs were passed through a nucleic acid extraction, using EazyMag (Biomerieux, QC, Canada) automatic total nucleic acid isolation protocol (500 µL UTM input, 70 µL elute).

The “classical” design of SAC for respiratory samples is publicly disclosed on the WHO web site (https://www.who.int/diagnostics_laboratory/eual/eul_0515_202_00_covid19_coronavirus_real_time_pcr_kit_ifu.pdf, accessed on 6 June 2021). The recommended SAC is based on the RNase P qPCR assay, used for measuring the number of human single copy genes, as an endogenous reference. More precisely, the assay detects the Ribonuclease P RNA component H1 (H1RNA) gene (*RPPH1*) on chromosome 14, cytoband 14q11.2. The assay location is chr.14:20343370 on build GRCh38. It has an 87 bp amplicon that maps within the single exon *RPPH1* gene. For clinical samples rich in bacterial content, the 16S rDNA qPCR assay was also described as a candidate for SAC (see [32] and references therein): the amplification of V3 and V4 variable regions of the eubacterial 16S rRNA gene (∼460 bp) was performed using 5′CCTACGGGNGGCWGCAG3′ and 5′GACTACHVGGGTATCTAATCC3′ primers, combined with TaqMan 16S probes [32].

All oligonucleotides used in this work were synthesized by Integrated DNA Technologies (IDT) (Iowa, IA, USA). The qPCR cycle threshold values (Cq) for each assay (presented in the Results and Discussion section, below) were extracted using the default parameters of the software, provided by the real-time PCR instrument manufacturer (LC 480 II Roche Software release 1.5.1.62sp2, QC, Canada). Amplification curves, obtained by the software, were visually verified to critically confirm positive and negative results.

## 3. Results and Discussion

Variability of the SAC signal: Each sample type (e.g., nasal, rectal or oral swabs, or saliva, or gargle) has its own frequency distribution of internal biomarker concentrations within the tested population. Based on this frequency distribution, a biomarker could be considered (or excluded) as a good candidate for SAC [2,15,33,34,35]. To illustrate the process in establishing a suitable SAC, we refer to our recent studies [1,3]. In this work, the quality of self-collected fecal material on rectal swabs (BBL CultureSwab, BD, ON, Canada) was analyzed, using two different biomarkers: bacterial 16S rDNA and human single copy gene (RNase P). Due to the much lower abundance of human DNA in the sample, bacterial DNA was a more reliable indicator of sample adequacy (Figure 1, left bell-curve). However, due to the multitude of bacterial genomic nucleic acids, the control-assay reaction could not be placed in the same compartment as the pathogen-specific assay. A one compartment reaction resulted in significant competitive inhibition due to the presence of excessive 16S rDNA targets. In contrast, when we used nasopharyngeal swabs (FloqSwabs, 503CS01.CA; Copan, VWR International, QC, Canada), the RNase P qPCR assay proved to be a well-suited candidate for SAC [1]. The Cq values of RNase P revealed a maximal frequency distribution characterized by the mean value of 27.2 Cq units, reflecting on average, a few thousand human genome equivalents (HGE) present in a “typical” clinical sample. Notably, the standard deviation of the Cq values for nasopharyngeal swabs is smaller than that for rectal swabs, reflecting an inherent difference in sampling variability. Once the biomarker which represents a “good” SAC is selected, the next task is to choose the quantitative values of SAC which would assure with reasonable confidence that a negative pathogen result is indeed negative. While analyzing for the presence of pathogen (influenza A virus) in nasopharyngeal samples, we observed that about 2% of swabs (out of 1821 tested) had less than 100 human genome equivalents and were never positive for influenza A virus [1]. We concluded that our confidence in a negative result is overestimated in these 2% of samples.

Test resulting via relative normalization: Current qPCR assays, including COVID-19 screening tests, are typically qualitative (positive/negative, i.e., binary resulting). Despite this, decision-based resulting algorithms contain quantitative rules. In qPCR-based screening tests, analytical sensitivity can reach a single-molecule detection. However, an analytically positive result does not necessarily imply that viable pathogen, and not molecular “debris”, is present in the sample. Conversely, an analytically negative result does not guarantee that pre-analytical steps were properly performed. In this regard, a lack of confidence in a declared positive and/or negative must be reported. This has led to the introduction of the term “indeterminate, or weak positive result”, as a clinical disclaimer for uncertainty in such cases. Strikingly, this new category of resulting (weak positives), during the current COVID-19 pandemic, approaches almost 40% of our day-to-day positive results (see also [12,13,16,17,18,26,27,28,36,37]).

Origin of “indeterminate or weak” positive results: Due to the need for simplified, binary reporting of test results, not all positive results are the same. COVID-19-positive nasopharyngeal swabs characterized by cycle threshold values in the range of 33 to 40 Cq units can be considered as “weak” positive. They could appear at the very early, initial stages of the infection when the viral load is just beginning to increase. Repeating a doubtful test result, with a second sample, will resolve this problem since the dynamics of infection will drive accumulation of the pathogen-specific biomarker. Weak positive results can, alternatively, be the result of pre-analytical factors which can be controlled by using SACs. However, the qPCR does not only detect viable pathogen, but also molecular debris, in the form of degraded nucleic acids. Therefore, analytically positive test results can be detected days after viable pathogen has been cleared from the patient’s body [14,18,20,38]. This is a further evidenced by data showing the inability to culture COVID-19 from low positive samples [14]. Labeling a patient as “infective”, in this context, will unnecessarily prolong their isolation and quarantine for days, sometimes weeks. Possessing additional information about the case history and a quantitative value for SAC could help with the final interpretation of the results. To reduce the fraction of weak positives and/or indeterminate results due to sampling variability, relative quantification between the 2 biomarkers might clearly indicate that a sample is “adequate”, or alternatively that sample quality and quantity has been compromised, resulting in a pathogen-negative or weak positive signal. Relative quantification is less dependent on external pre-analytical variables as compared to absolute values (individual Cq values) which are currently in use by a number of authorized COVID-19 assays. Additionally, the relative quantification (using SAC as a biomarker) allows different technological platforms to be analyzed and compared in parallel [35].

Although the importance (and added value) of using SAC is clear, there is a technical caveat in incorporating SAC into routine testing. The presence of SAC in the same reaction mix could “attenuate” a weak pathogen-positive signal due to its competitive nature. This problem can be eliminated by (1) performing the SAC assay in a separated reaction compartment/chamber or (2) developing an approach to overcome competition within a single reaction. To allow for a single compartment assay, we engineered PCR assays with reduced efficiency compared to that of the pathogen-specific qPCR reaction. This concept of non-competitive SAC has immediate applicability for infection control screening tests.

The integral, non-competitive SAC: Further in our quality assurance work, we illustrate the use of SAC which is suitable for a single reaction compartment (Table 1 and Table 2). This control does not compete and therefore does not compromise the detection of weak pathogen-positive samples, which contain low concentrations of pathogen genetic material. To achieve this goal, assay conditions were engineered to artificially decrease the efficiency of the SAC qPCR reaction and keep it well controlled. It is well-known that primer design influences qPCR efficacy [39,40,41,42,43]. Therefore, using lower PCR efficacy, one can enforce sample-specific biomarker “expression” only if the sample is negative for a pathogen-specific biomarker.

The following example illustrates this strategy of non-competitive SAC assay design: annealing/priming of RNase P is performed as previously described [1,3]. The reverse primer and TaqMan probe are common to all assays presented on Table 2. The classical RNase P forward primer (5′AGATTTGGACCTGCAGCG3′) has an annealing temperature of Tm ~ 65 °C, compatible with the actual annealing temperature of RNase P qPCR assay (set at Tm—5 °C, i.e., at 65°–50° = 60 °C). The re-designed (non-competitive) primer(s) has Tm ~ 2–5 °C below the imposed annealing/extension temperature of the qPCR assay (60 °C). Such primer design deviates about 8–10 °C from the recommended melting temperatures [41,42,44,45]. This unique thermal (in)stability of one of the primers permits the programed non-competitive nature of the reaction, resulting in assay sensitivity drop and change in the limits of detection (LoD) toward 100 (Table 1, column #A, Tm = 58 °C), or even 1000 human genome equivalents (Table 1, column #B, Tm = 55 °C) of human DNA. The exact primer sequences of each SAC assay are presented in Table 2. Concerning the non-competitive SAC assay, the target gene present in each sample (RNase P) is multiplied at a lower amplification rate, defined by melting profile (Tm) of the forward primer in use (Table 2).

The criteria for choosing a SAC assay should depend on the quantitative distribution of SAC in the sampling population, density of disease-specific vs. sample specific biomarkers and on the epidemiological needs for the confidence in a negative result. Considering that less than 2% of all negative nasopharyngeal swabs have Cq values of RNAse P larger than 33 Cq units (classical RNase P assay), reporting “inconclusive” results for these “negative” samples will generate additional value for future infection control efforts.

## 4. Conclusions

Herein we indicated that SAC has important analytical, clinical, and epidemiological value. SAC not only increases the clinical and epidemiological confidence in a negative test result by serving as an integral qPCR assay control but also ensures that all up-stream steps were adequate. We emphasize the universal value of SAC and describe a simple technical strategy to introduce non-competitive SAC which does not interfere with the limit of detection for the screened pathogen. Incorporation of SAC will not only facilitate the future pandemic management but will also bring more systematic knowledge into quantitative sampling variability and its impact on quantitative molecular diagnostics.

## Figures and Tables

**Figure 1 diagnostics-11-01133-f001:**
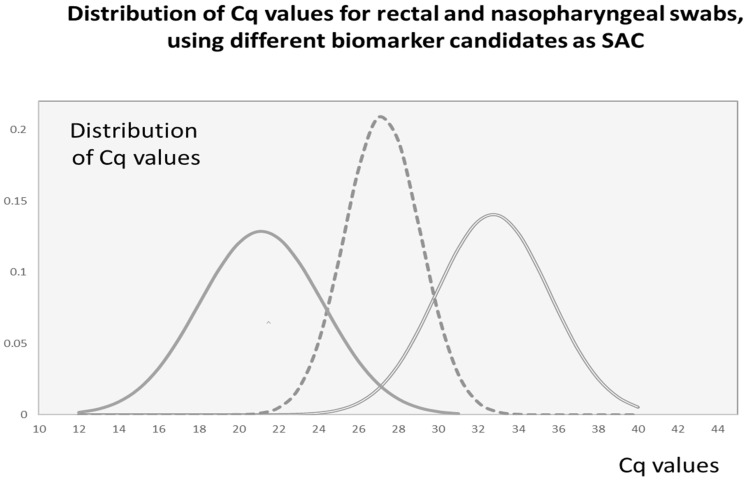
Re-plotting frequency distributions of Cq values for rectal and nasopharyngeal swabs, using different biomarkers as candidates for SAC [1,3]; *X*-axis: reported Cq values; *Y*-axis: frequency distribution of Cq values, using experimentally measured average and standard deviation (SD) of Cq values (total number of data = N) for each set of samples (rectal or nasopharyngeal swab) and each biomarker (16S rDNA or RNase P) as reported [1,3]. Legend: Rectal swab (left and right bell curves): The distribution of Cq values of 16S rDNA (reflecting bacterial load) is presented on left curve (solid line, (-)), average Cq value = 21.13, SD = 3.1, N = 713; and Cq’s of RNase P is on the far right, (double line, (=)), average Cq = 32.74, SD = 2.8, N = 527 (note that 26.1% of self-collected rectal swabs did not generate an RNase P Cq value); Nasopharyngeal swab (middle bell curve with intermittent line, (--)): Average Cq value = 27.2 (~10,000 GE), SD = 1.9 Cq units, N = 1821. Three SD units from the mean Cq value toward the right side of the scale (27.2 + (3 × 1.9)) will produce a Cq = 32.9. Data with RNase P Cq < 33 was present in >98% of samples, indicating that pathogen-negative results with RNase P Cq ≥ 33 should be repeated if possible.

**Table 1 diagnostics-11-01133-t001:** Relation between Cq values of classical SAC assay and non-competitive SAC assay: note a shift of qPCR product appearance (of non-competitive SAC) toward high Cq values. This is allowing detection of low pathogen-positive signals, and offering confidence in negative result.

		Average (*n* = 3)	
Number of Human GE	Cq (Classical)	Cq (Non-Competitive #A)	Cq (Non-Competitive #B)
10,000–100,000	25.3	29.2	33.2
1000–10,000	28.7	32.4	35.5
100–1000	31.5	36.1	37.5
10–100	35.2	39.4	n.d.
1–10	38.1	n.d.	n.d.

Performance difference between classical (RNase P-based) SAC assay [1,3,12,13,36] and non-competitive SAC assays; Legend: The RNase P assay is performed as described (see [1] and referenced work). The calibration between human genome equivalents and Cq values is estimated using dilutions of known human genomic DNA standards.

**Table 2 diagnostics-11-01133-t002:** SAC assay (RNase P) with “classical” and “non-competitive” primer design of forward primer, allowing for the design of a duplex assay reaction, whereby priority in PCR efficacy will be “given” to the pathogen-specific signal.

Ribonuclease P (SAC)	Classical Assay	Non-Competitive #A	Non-Competitive #B
Forward primer	5′AGATTTGGACCTGCGAGCG3′	5’GGACCTGCGAGCG3’	5’GACCTGCGAGCG3’
Reverse primer	5′GAGCGGCTGTCTCCACCAGT3′		
TaqMan probe	5′FAMTTCTGACCTGAAGGCTCTGCGCG3′BQH		

The classical SAC assay design can outcompete low pathogen-positive samples (30 < Cq < 40). The non-competitive SAC assays (#A and #B) minimize competition with the low pathogen-positive signal. The melting temperature (Tm) of the forward RNase P primers [1,3] was changed from the original Tm = 64.6 °C (Table 1, column: classical assay) to Tm = 58.1 °C (Table 1, middle column #A), or Tm = 54.2 °C (Table 1, right column #B). Regarding the non-competitive SAC assays, the target undergoes cycles at a slower amplification rate. The described primer design allows for the programed non-competitive nature of the reaction; sensitivity drop and change in the limits of detection (LoD) toward 100 (#A) or even 1000 genome equivalents (#B) of human DNA.

## Data Availability

Not applicable.

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
