# Peer review of "Sample Adequacy Control (SAC) Lowers False Negatives and Increases the Quality of Screening: Introduction of “Non-Competitive” SAC for qPCR Assays"

_diagnostics, 2021, doi:10.3390/diagnostics11071133_

Round 1

Reviewer 1 Report

I find that sometimes the message is unnecessarily complicated.

Below, I propose some simplifications of the text, as an example

if the authors want accept and pursue along similar lines:

simpler tittle:

Non-competitive qPCR assay:

Sample Adequacy Control lowers false negatives and increases the quality of screening

 Text of abstract and so on :

Abstract: Sample Adequacy Control (SAC) has important/significant analytical, clinical, and epidemiological value. The SAC is an integral qPCR assay control, ensuring that all downstream preanalytical and analytical steps are adequate for testing. It increases the clinical and epidemiological confidence in a negative test result. A negative SAC and negative result on the pathogen screen/detection indicate that the preanalytical steps are not optimal, and that the test outcome should be declared  is inconclusive. Despite this, many regulatory-approved tests do not incorporate SAC in their assay design. Furthermore, the 21 regulatory approval of screening tests without SAC should not be accepted as a long-term strategy due to the attenuation of epidemiological efforts in tracing, isolating, and quarantining pathogen-positive individuals. Herein, we emphasize the universal value of SAC and offer, for the first time, a simple technical strategy to introduce non-competitive SAC, which does not interfere with the limit of detection for the screened pathogen.

INTRODUCTION

Sample adequacy control (SAC) is a “built-in check” to ensure that the input sample is suitable for testing [1-6]. Usually, the endogenous, sample-specific (human) gene is co-amplified to assure that an adequate sample has been collected.

 Define criteria of “adequacy” of a sample to justify the procedure you propose.

Author Response

Many thanks for suggesting changes which (indeed) are helping to make our work more readable and clearer: the whole manuscript (title: Sample Adequacy Control lowers false negatives and increases the quality of screening: use of “non-competitive” qPCR assay) is intensely modified with a full integration of comments from all authors, a suggested. We also want to thank review for the specific suggestion related to the clarification of “criteria of Sample Adequacy Control’. We added a full paragraph in Introduction:

“In this work, we emphasize the importance of sample adequacy in medical diagnostics. Sample adequacy is characterized by sufficient quantity and integrity of disease-specific biomarker present in the sample. One approach to establishing a reliable indicator of adequacy is to examine the frequency distribution of the sample-specific biomarker(s) in advance. This statistical approach is arbitrary but demonstrates that not all samples are equal in their biomarker quantity. For example, if a sample-specific biomarker quantity is found to be less than two standard deviations from its mean frequency distribution for a particular sample type, then this sample belongs to a small group of only 2.5% of samples. The rest of the samples (97.5%) contain higher biomarker quantity. The second approach is pathogen/disease specific. In this method, the lowest ratio of disease-specific biomarker versus sample-specific biomarker must be determined. For example, we never detected nasopharyngeal samples having an Influenza A virus density lower than 0.0001 per cell (i.e. one genome-equivalent of Influenza A virus in the context of 10 000 human genome equivalents) [1]. To achieve a maximal clinical/epidemiological sensitivity, samples should never have less than 10 000 human genome equivalents. When the sample contains less than 10 000 human genome equivalents, the risk of obtaining a false negative result when screening for a pathogen significantly increases. Since the concept of SAC is based on the frequency distributions of biomarkers, choosing a quantitative “threshold” for a sample-specific biomarker might de-pend on the gravity of the disease (Ebola versus simple respiratory infection) and the necessity of not missing a case…”

As well as in the Results and discussion segment:

“The criteria for choosing a SAC assay should depend on the quantitative distribution of SAC in the sampling population, density of disease-specific versus sample-specific biomarkers, and on the epidemiological needs for the confidence in a negative result. Considering that less than 2% of all negative nasopharyngeal swabs have Cq values of RNAse P larger than 33 Cq units (classical RNase P assay), reporting “inconclusive” results for these “negative” samples will generate additional value for future infection control efforts. “

Best regards, on the behalf of all authors, Ivan Brukner

Reviewer 2 Report

This timely paper points out to multiple benefits of SAC and demonstrates that a significant percent of negative reports should be declared failed tests if SAC is performed.

Newly developed non-competitive SAC provides simple and cost effective solution for implementing SAC without affecting sensitivity of the test. 

Publishing this paper will provide further motivation for a mandatory implementation of SAC.

Author Response

Many thanks for recognizing the importance of this work and its impact on future pandemic situations. Also, we appreciate the reviewer’s comments: the whole manuscript (title: Sample Adequacy Control lowers false negatives and increases the quality of screening: use of “non-competitive” qPCR assay) is intensely modified with full integration of comments from all authors, as suggested.

Best regards, on the behalf of all authors, Ivan Brukner
